# Vi2ACT:Video-enhanced Cross-modal Co-learning with Representation Conditional Discriminator for Few-shot Human Activity Recognition

Kang Xia
Nanjing University
Nanjing, China
xiakang@smail.nju.edu.cn

Wenzhong Li*
Nanjing University
Nanjing, China
lwz@nju.edu.cn

Yimiao Shao
Nanjing University
Nanjing, China
yimiao_shao@smail.nju.edu.cn

Sanglu Lu
Nanjing University
Nanjing, China
sanglu@nju.edu.cn

## ABSTRACT

Human Activity Recognition (HAR) as an emerging research field has attracted widespread academic attention due to its wide range of practical applications in areas such as healthcare, environmental monitoring, and sports training. Given the high cost of annotating sensor data, many unsupervised and semi-supervised methods have been applied to HAR to alleviate the problem of limited data. In this paper, we propose a novel video-enhanced cross-modal collaborative learning method, Vi2ACT, to address the issue of few-shot HAR. We introduce a new data augmentation approach that utilizes a text-to-video generation model to generate class-related videos. Subsequently, a large quantity of video semantic representations are obtained through fine-tuning the video encoder for cross-modal co-learning. Furthermore, to effectively align video semantic representations and time series representations, we enhance HAR at the representation-level using conditional Generative Adversarial Nets (cGAN). We design a novel Representation Conditional Discriminator that is trained to assess samples as originating from video representations rather than those generated by the time series encoder as accurately as possible. We conduct extensive experiments on four commonly used HAR datasets. The experimental results demonstrate that our method outperforms other baseline models in all few-shot scenarios.

## CCS CONCEPTS

• **Human-centered computing** → **Ubiquitous and mobile computing systems and tools**; • **Computing methodologies** → **Machine learning algorithms**.

*The corresponding author is Wenzhong Li.

## KEYWORDS

Human activity recognition, Few-shot Learning, Conditional Generative Adversarial Nets

**ACM Reference Format:**
Kang Xia, Wenzhong Li, Yimiao Shao, and Sanglu Lu. 2024. Vi2ACT:Video-enhanced Cross-modal Co-learning with Representation Conditional Discriminator for Few-shot Human Activity Recognition. In *Proceedings of the 32nd ACM International Conference on Multimedia (MM '24), October 28-November 1, 2024, Melbourne, VIC, AustraliaProceedings of the 32nd ACM International Conference on Multimedia (MM'24), October 28-November 1, 2024, Melbourne, Australia.* ACM, New York, NY, USA, 9 pages. https://doi.org/10.1145/3664647.3681664

## 1 INTRODUCTION

Human Activity Recognition (HAR), as an emerging research field, has garnered widespread academic attention [28, 40, 42]. HAR aims to analyze and understand individual or group behaviors, habits and other behavioral characteristics. With the rapid development of smartphones [16, 27] and wearable devices, sensor technology has been widely employed. The diverse and rich data generated by sensors, such as accelerometers [12, 42], gyroscopes, and magnetometers [21], provide detailed behavioral information, enabling researchers to comprehensively and precisely unveil human behavioral patterns in various environments. Its applications span across various domains, including healthcare [16], smart home [13], environmental monitoring [5], among others.

The development of HAR has undergone several stages, initially focusing on manual representation extraction. It later entered the era of machine learning, employing traditional algorithms such as support vector machines and decision trees. In recent years, with the rise of deep learning technology, especially the successful application of convolutional neural networks (CNN) [28, 30, 31], recurrent neural networks (RNN) [4, 22] and Transformer [37, 40], HAR has achieved end-to-end representation learning, greatly improving the accuracy of recognition.

In deep learning-based HAR, a significant drawback is the cost and complexity associated with acquiring labeled data. Training supervised learning models requires a substantial amount of labeled data, and manually annotating datasets is a time-consuming and expensive task.

Recently, Xia et al. [40] proposed TS2ACT to achieve few-shot HAR. It uses label text to search for human activity images via a search engine. Then it adopts a pre-trained CLIP [29] image encoder to jointly train with a time series encoder using contrastive learning, where the time series and images are brought closer in representation space if they belong to the same activity class. Although using image representations for contrastive learning can effectively enhance time series representations, it is still challenging to distinguish temporally opposite activities such as "opening a door" and "closing a door" due to the lack of temporal information in images. Moreover, the lack of a one-to-one correspondence between time series data and their labeled image classes means that straightforward application of contrastive learning may lead to inaccuracies in data alignment.

To address the aforementioned issues, we propose a novel video-enhanced cross-modal co-learning method, Vi2ACT, to achieve few-shot HAR. We propose utilizing video semantic representations to enhance the time series encoder instead of image semantic representations. Initially, we employ a text-to-video generation model to create label-related videos, thereby generating semantically richer video content. Subsequently, we use these videos along with their corresponding labels to train a video encoder. Second, we design a novel Representation Conditional Discriminator. This module is trained to assess samples as originating from video encoder rather than those generated by the time series encoder. Through the adversarial training, action representations and video semantic representations can be aligned more efficiently.

To summarize, our main contributions are as follows.

- We propose a novel cross-modal co-learning approach Vi2ACT to strengthen few-shot HAR. To obtain a large number of video semantic representations, we introduce a novel data augmentation approach that leverages a text-to-video generation model to generate label-related videos. Subsequently, we fine-tuning the video encoder for cross-modal co-learning.
- We design a novel Representation Conditional Discriminator that guides the time series encoder to extract class-specific semantic representations through specific class conditions, thus improving HAR performance. To our knowledge, this is the first attempt to enhance HAR from the representation level using conditional Generative Adversarial Nets.

## 2 RELATED WORK

### 2.1 Supervised Human Activity Recognition

Early HAR research primarily focused on manual representation extraction and traditional machine learning techniques such as SVM [8] and KNN [33]. Consequently, it often required domain expertise and was limited to capturing surface-level features.

In contrast, deep learning models have made significant strides in autonomously learning hierarchical representations from raw sensor data. Among the various deep learning architectures, CNNs [25, 28], RNNs [4, 22] and Transformers-based [9, 34] method have exhibited promising performance. For instance, Zeng et al. [43] design a CNN-based HAR method to capture the local correlations and scale invariance of signals. Challa et al. [4] propose a CNN-BiLSTM

model which enables learning of both local representations in sequential data and long-term dependency relationships. Conformer-HAR [17] introduces the Conformer, which is the state-of-the-art (SOTA) model in the field of speech recognition.

### 2.2 Semi/Un-supervised and Few-shot HAR

In order to alleviate the problem of scarcity of labeled data, semi-supervised, unsupervised learning, and particularly few-shot learning methods have been widely used in HAR.

Semi-supervised HAR [3, 23] primarily utilizes labeled data to train an encoder, which is then used to assign pseudo-labels to a large amount of unlabeled data for further encoder training. To tackle the constraints posed by limited labeled data and class imbalance, Chen et al. [6] propose a pattern-balanced semi-supervised method. Tang et al. [36] utilize confidence to identify the most relevant samples and propagates labels to the most confident samples.

Unsupervised HAR [11, 41] typically employs a two-stage pipeline, involving unsupervised pretraining followed by fine-tuning on labeled data. ColloSSL [14] leverages natural transformations in the sensor data from multiple devices to perform contrastive learning, and learns a robust encoder. Ma et al. [24] propose a multi-task learning framework integrating CNN-BiLSTM autoencoder, K-means clustering, and classification tasks to achieve unsupervised HAR.

Few-shot HAR methods aim to recognize novel activities with only a few or even one labeled sample. TS2ACT [40] uses label text to search for human activity images, and then jointly trains a time series encoder with the CLIP [29] image encoder to achieve few-shot HAR. RF-CM [39] proposes a general cross-modal framework which achieves few-shot radar-based HAR by using a large number of public WiFi data.

### 2.3 Cross-modal HAR and LLMs for HAR

Due to the high cost associated with collecting sensor data, recent studies have utilized vision as an auxiliary modality to assist models in extracting representation from sensor data. For instance, IMU-Tube [18, 19] proposes to generate virtual sensor data from videos by applying pose estimation techniques to infer the 3D motion of the human body in the video. This approach typically relies on starting with a large number of videos, as only a segment of IMU signal can be extracted from a single video. Additionally, since this method often depends on the accuracy of pose estimation algorithms, it makes the method highly sensitive to occlusions of the human body. In contrast, our method does not require pose estimation; instead, it utilizes a video action recognition model to directly learn embeddings for HAR from entire video segments.

In addition, large language models (LLM) have been used to achieve zero-shot HAR. HealthLearner [20] explores the potential of LLMs as general few-shot learners based on time-series health data. They discover that with minimal in-context tuning, LLMs could ground digital time-series data from wearable and clinical-grade sensor devices , significantly improving zero-shot inference and supervised baselines in tasks such as activity recognition, calorie computation, and atrial fibrillation classification. HARGPT [15] processes raw IMU data by inputting it into an LLM and uses role-playing and "think step-by-step" strategies for prompting, successfully identifying human activities from the raw IMU data.

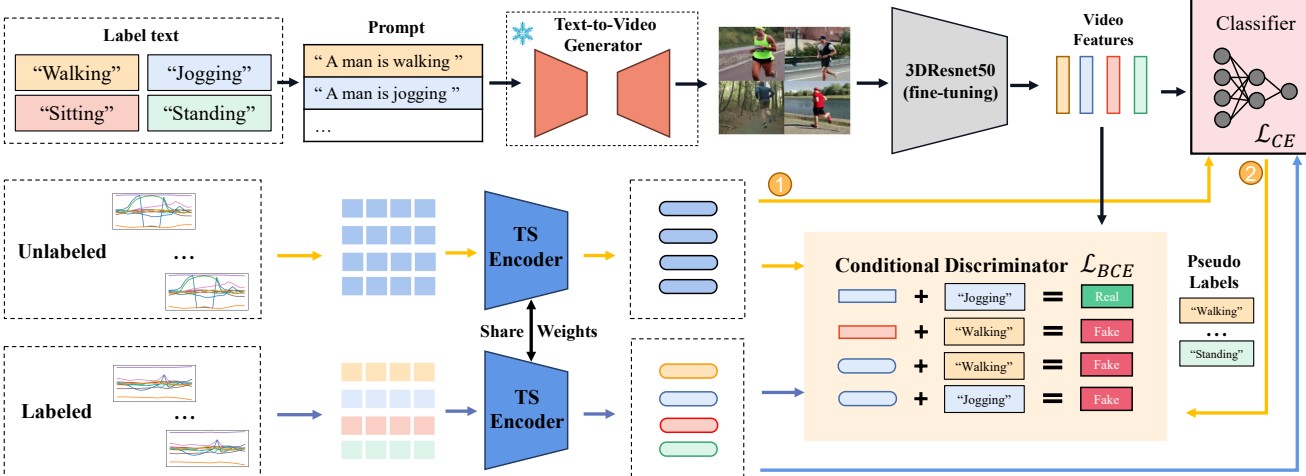

**Figure 1: The framework of Vi2ACT.**

## 3 METHOD

### 3.1 Overall Architecture

As shown in Figure 1, the overall architecture of Vi2ACT comprises five main components: a pre-trained text-to-video generator, a video encoder $\mathcal{F}(\cdot)$, a representation classifier $g(\cdot)$, a time series encoder $G(\cdot)$, and the Representation Conditional Discriminator $D(\cdot, \cdot)$. Initially, we input label text into prompt templates as the input for the text-to-video generator to generate a substantial amount of label-relevant videos. Subsequently, these videos are utilized to train the video encoder $\mathcal{F}$ and the representation classifier $g$. Then the time series encoder $G$ is employed to obtain representation from the input time series and the Representation Conditional Discriminator $D$ works by discerning whether the representations originate from time series encoder $G$ or video encoder $\mathcal{F}$, facilitating the joint optimization of the time series encoder. Finally, we employ the representation classifier $g$ to directly classify the time series representations.

We denote the collection of sensor data samples as $S = \{s_1, s_2, ...s_n\}$, where each $s_i$ is derived by segmenting the original signal into fixed-size segments. Each sample $s_i \in \mathbb{R}^{L \times D}$ is characterized by a multivariate time series, where $L$ represents the length of the sample, and D is the channel dimension, containing all x, y, and z-axis from the accelerometer, gyroscope, and magnetometer.

### 3.2 Cross-modal Video representation Generation

Due to the limited availability of labeled data, we propose to use text-guided video generation to supplement the data deficiency. Initially, we fill the label text (e.g., "jogging") into pre-defined prompt templates (e.g., "A man is [class]") to construct textual prompts (e.g., "A man is jogging"). Subsequently, these prompts are fed into a text-to-video generation model to create videos relevant to the labels. We utilize an open-source video generation model ModelScopeT2V [38] to generate large amounts of semantically richer

videos. Following this, we employ a pre-trained video encoder to acquire all the video representations.

Suppose that video $X_c$ is a **randomly selected** video from the videos relevant to the class $c$. We use the video encoder $\mathcal{F}(\cdot)$ to obtain video representations $f_c^V \in \mathbb{R}^d$ by

$$f_c^V = \mathcal{F}(X_c) \tag{1}$$

### 3.3 Time Series Encoder

The time series encoder (denoted by $G(\cdot)$) is intended for recognizing and encoding correlations within sensor data at various time steps. However, unlike words in a sentence, individual time steps in time series data lack inherent semantic meaning. Therefore, extracting local semantic information becomes crucial for meaningful analysis. Previous HAR methods often directly applied Transformer to time series data. In contrast, we enhance locality and capture subsequence-level semantic information by aggregating time into subsequences, as shown in Figure 2(a).

Specifically, for an input time series data $s \in \mathbb{R}^{L \times D}$, we first segment each input into patches $\text{Patching}(s) \in \mathbb{R}^{N \times l \times D}$ of equal size using a sliding window of width $l$ with a 50% overlap, where $\text{Patching}(\cdot)$ denotes the patching operation and $N$ is the number of patches. To address any shortfall in the last patch, we duplicate the last value and pad it to the end of the original sequence. Then the input data dimensions are mapped to a $d$-dimensional Transformer latent space $s^p \in \mathbb{R}^{N \times l \times d}$ through a convolutional backbone consisting of 1D-CNN layers and SiLU [10] activation functions.

$$s^p = \text{SiLU}(\text{CNN}(\text{Patching}(s))) \tag{2}$$

We employ a standard Transformer [37] as our backbone network, consisting of several Transformer blocks. We take the average along the length dimension of $s^p \in \mathbb{R}^{N \times l \times d}$ and adding learnable positional encodings $\mathbf{W}_{pos} \in \mathbb{R}^{N \times d}$ to get the input $s^d = \text{Avg}(s^p) + \mathbf{W}_{pos}$, where $s^d \in \mathbb{R}^{N \times d}$ denote the token which

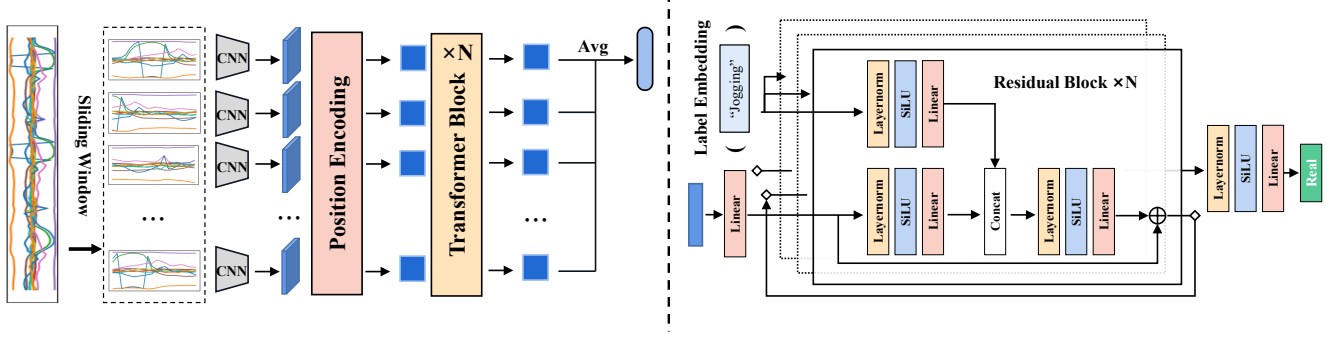

(a) Time Series Encoder

(b) Representation Conditional Discriminator

Figure 2: The framework of Time Series Encoder and Representation Conditional Discriminator.

will be fed into Transformer encoder. Positional encoding ensures that each position in the input sequence has a unique identifier, enabling the Transformer model to utilize attention mechanisms for processing information from different positions. Then the stacked Transformer blocks transform the input into hidden layer outputs through a variety of modules, operating as follows:

$$
\begin{aligned}
AttentionOutput_t &= \text{Self-Attention}(s_{t-1}^d) \\
LayerNormOutput_t &= \text{LayerNorm1}(s_{t-1}^d + AttentionOutput_t) \\
FFNOutput_t &= \text{FFN}(LayerNormOutput_t) \\
s_t^d &= \text{LayerNorm2}(LayerNormOutput_t + FFNOutput_t)
\end{aligned}
\tag{3}
$$

Here, Self-Attention($\cdot$) denotes multi-head self-attention, which efficiently simulates interdependencies between positions in the time series, capturing critical temporal relationships. The Feedforward Neural Network FFN($\cdot$) enhances model expressiveness by introducing non-linearity to better represent inputs at a higher level. The normalization operation LayerNorm($\cdot$) helps alleviate the issue of internal covariate shift, enhancing independence between representations, thereby improving the robustness and training efficiency of the network.

Finally, we take the average of the output from the last Transformer block $z^d \in \mathbb{R}^{N \times d}$ as the input time series representation $f^T \in \mathbb{R}^d$. Therefore, for an input $s \in \mathbb{R}^{L \times D}$, we can obtain its representation through the encoder $G(\cdot)$ by.

$$
f^T = G(s) \in \mathbb{R}^d.
\tag{4}
$$

### 3.4 Representation Conditional Discriminator

To enhance the capability of the time series encoder in capturing richer semantic representations, we align video representations and time series representations within the same representation space. However, simply using Contrastive Learning [7] or Mean Squared Error loss may lead to inaccuracies in model alignment. To address this issue, we introduce a Representation Conditional Discriminator (denoted by $D(\cdot, \cdot)$) to implicitly align video representations and time series representations.

The Time Series encoder and the Representation Conditional Discriminator respectively play the roles of the generator model $G$ and the discriminator model $D$ in the Conditional Generative Adversarial Network (cGAN). cGAN [26] is a method for training generative models. The generator $G$ is trained to produce outputs that the discriminator $D$ cannot distinguish from "real" data. The discriminator $D$ is trained to evaluate samples as accurately as possible, determining whether they originate from the training data or are generated by $G$. In our framework, the Time Series encoder is utilized to map $s_c$ to $f_c^T$, where $s_c$ represents a time series sample belonging to the action category $c$. The discriminator $D(f, c)$ outputs a scalar indicating the probability that representation $f$, under the condition of class $c$, originates from the video rather than $G$. The Time Series encoder $G(\cdot)$ learns to deceive the discriminator. The discriminator $D$ is trained to classify between fake tuples $\{f_c^T, c\}$, $\{f_i^V, c | \text{where } i \neq c\}$ and real tuples $\{f_c^V, c\}$, as shown in Figure 1. Both the generator $G$ and the discriminator $D$ are trained simultaneously. Specifically, the parameters of $G$ are adjusted to minimize $\log(1 - D(G(s_c), c))$, while the parameters of $D$ are tuned to minimize $\log D(f_c^V, c)$ and $\log(1 - D(f_i^V, c))$. The objective of the Representation Conditional Discriminator can be expressed as:

$$
\begin{aligned}
\min_G \max_D \mathcal{L}_{cGAN}(G, D) = &\mathbb{E}_{f_c^V, c} \left[ \log D(f_c^V, c) \right] + \\
&\mathbb{E}_{f_i^V, c} \left[ \log(1 - D(f_i^V, c)) \right] + \\
&\mathbb{E}_{s_c, c} \left[ \log(1 - D(G(s_c), c)) \right]
\end{aligned}
\tag{5}
$$

Representation Conditional Discriminator adopts a fully connected network with multiple residual blocks as its backbone, as shown in Figure 2(b). Layer normalization is applied to the discriminated representations within each residual block. To distinguish representation categories within discriminator, class-specific condition $c$ projections are introduced in each residual block. The inference process of the discriminator at block $t$ is shown in Equation 6

$$
\begin{aligned}
C &= W_t^{(i)}(\text{SiLU}(\text{Layernorm}(c))) + b_t^{(i)} \\
F &= W_t^{(j)}(\text{SiLU}(\text{Layernorm}(f_{t-1}))) + b_t^{(j)} \\
f_t &= W_t^{(k)}(\text{SiLU}(\text{Layernorm}(\text{ConCat}(F, C)))) + b_t^{(k)}
\end{aligned}
\tag{6}
$$

Ultimately, the output of the last block undergoes LayerNorm [1], SiLU [10], and a linear layer to serve as the final output, i.e., $D(f, c)$.

## 3.5 Representation Classifier

After extracting the representation of the time series using the encoder, we employ a representation classifier $g(\cdot)$ to derive the final classification results. The representation classifier comprises three fully connected neural network layers with SiLU activation and is trained with labeled samples. Assuming $\boldsymbol{y}$ represents the label, and $\hat{\boldsymbol{y}} = g(f)$ is the probability distributions predicted by the classifier, the training objective of the representation classifier is to minimize the classification error by employing cross-entropy loss.

$$\mathcal{L}_{CE} = -\sum_i \boldsymbol{y}_i \log(\hat{\boldsymbol{y}}_i) \tag{7}$$

## 3.6 Train the Vi2ACT model

*3.6.1 Fine-tuning the Video Encoder.* Although a pre-trained video encoder can effectively extract action representations from videos, directly using it for video representation extraction may result in suboptimal performance due to significant disparities between the pre-trained and generated videos. Therefore, we initially fine-tune the pre-trained video encoder using the generated videos. Simultaneously, we optimize the above Representation classifier, employed as the video encoder's classifier, through cross-entropy loss. This approach eliminates the need to train a representation classifier from scratch while effectively classifying semantic representations. In particular, for the generated video $X_i$, we use the following loss function for fine-tuning.

For the generated video segments $f_c^V = \mathcal{F}(X_c)$, we utilize cross-entropy loss to simultaneously optimize the video encoder and representation classifier through the following formula.

$$\mathcal{L}_{CE} = -\sum_i \boldsymbol{y}_i \log(g(\mathcal{F}(X_i))) \tag{8}$$

*3.6.2 Training the Time Series Encoder and Representation Conditional Discriminator.* The encoder and discriminator are trained following the Conditional Generative Adversarial Network (cGAN) framework. For unlabeled data, we first use the time series encoder $G$ to extract time series representations, which are then fed into a classification head to obtain pseudo-labels (e.g., $c'$) for the unlabeled time series (e.g., $s_{c'}$). To jointly leverage labeled and unlabeled data while mitigating the impact of noisy pseudo-labels, we utilize the following loss function to optimize the time series encoder and Representation Conditional Discriminator.

$$
\begin{aligned}
\min_G \max_D \mathcal{L}_{cGAN}(G, D) =& \mathbb{E}_{f_c^V, c}\left[\log D(f_c^V, c)\right] + \\
& \mathbb{E}_{f_i^V, c}\left[\log(1 - D(f_i^V, c))\right] + \\
& \mathbb{E}_{s_c, c}\left[\log(1 - D(G(s_c), c))\right] + \\
& \lambda\mathbb{E}_{s_{c'}, c'}\left[\log(1 - D(G(s_{c'}), c'))\right]
\end{aligned}
\tag{9}
$$

Here, the hyperparameter $\lambda \in (0, 1)$ is employed to achieve a balance between labeled and unlabeled data during training.

**Table 1: Statistics of the datasets. The four columns represent the number of users, the number of activities, the sensor type (A = accelerometer, G = gyroscope, M = magnetometer)**

| Dataset | Users | Activity classes | Sensor Type |
| --- | --- | --- | --- |
| UCI-HAR | 30 | 6 | A,G |
| HHAR | 9 | 6 | A,G |
| MotionSense | 24 | 6 | A,G |
| PAMAP2 | 9 | 12 | A,G,M |

## 4 EVALUATION

In this section, we extensively evaluate the proposed model based on several publicly available HAR datasets.

## 4.1 Datasets

We consider four publicly available datasets that cover a wide variety of device types and activity recognition tasks in different environments. Table 1 provides a summary of the statistical information for all four datasets. For data preprocessing, we only use the acceleration, gyroscope and magnetometer sensing signals to form the time series. Then we split the time series into segments of equal window size as input. Each segment contains 500 data points with 50% overlap. Following the literature [32, 40], we randomly selecting 80% of the users to construct the training set, and the rest for testing. The validation data is the unlabeled data in the training set, which is used for generating pseudo-labels and fine-tuning the model hyperparameters.

## 4.2 Baseline Algorithms

We compare the proposed method with several state-of-the-art HAR solutions including both supervised and semi-supervised methods. The following are two fully supervised baseline algorithms.

- **TCN** [2]. TCN combines the best practices such as dilated convolutions and residual connections with causal convolutions for autoregressive prediction.
- **ConformerHAR** [17]. ConformerHAR introduces the state-of-the-art (SOTA) model Conformer in the field of speech recognition. Furthermore, they improved the performance by incorporating CNN layers that excel at extracting local representations effectively.

We also compare our method with mainstream semi-supervised and unsupervised HAR algorithms as follows.

- **FixMatch** [35]. FixMatch is a semi-supervised learning framework. It generates pseudo-labels for large amounts of unlabeled data and then use these pseudo-labels, along with the original labeled data, to train the model.
- **SimCLR** [7]. The idea of SimCLR is to learn useful features through contrastive learning. The algorithm augments input data with random transformations (such as flipping, cropping, rotating, etc.) and uses a shared neural network to extract features. It then uses a contrastive loss function to encourage features between similar classes to be closer together, and features between dissimilar classes to be farther

**Table 2: Performance comparison with different HAR methods. In the table, "*" indicates a supervised learning algorithm, "+" indicates a semi-supervised learning algorithm, "-" indicates an unsupervised learning algorithm.**

| Dataset | | UCI-HAR | | | HHAR | | | MotionSense | | | PAMAP2 | | |
|---|---|---|---|---|---|---|---|---|---|---|---|---|---|
| Model | Shot | Acc | Recall | F1 | Acc | Recall | F1 | Acc | Recall | F1 | Acc | Recall | F1 |
| * TCN | 1 | 0.433 | 0.445 | 0.382 | 0.324 | 0.315 | 0.309 | 0.387 | 0.362 | 0.352 | 0.358 | 0.320 | 0.309 |
| * ConformerHAR | 1 | 0.488 | 0.465 | 0.423 | 0.364 | 0.366 | 0.351 | 0.455 | 0.422 | 0.413 | 0.363 | 0.330 | 0.319 |
| ¯ SimCLR | 1 | 0.606 | 0.577 | 0.633 | 0.506 | 0.498 | 0.488 | 0.563 | 0.561 | 0.553 | 0.469 | 0.446 | 0.433 |
| ¯ MDC | 1 | 0.729 | 0.713 | 0.719 | 0.682 | 0.682 | 0.680 | 0.727 | 0.711 | 0.709 | 0.641 | 0.624 | 0.622 |
| + FixMatch | 1 | 0.629 | 0.630 | 0.620 | 0.526 | 0.540 | 0.512 | 0.582 | 0.551 | 0.533 | 0.566 | 0.542 | 0.538 |
| + TS2ACT | 1 | 0.758 | 0.757 | 0.758 | 0.711 | 0.701 | 0.703 | 0.747 | 0.744 | 0.743 | 0.692 | 0.697 | 0.676 |
| Ours | 1 | **0.782** | **0.781** | **0.780** | **0.755** | **0.754** | **0.743** | **0.761** | **0.753** | **0.750** | **0.733** | **0.725** | **0.721** |
| * TCN | 5 | 0.567 | 0.566 | 0.493 | 0.432 | 0.422 | 0.416 | 0.472 | 0.430 | 0.421 | 0.520 | 0.468 | 0.471 |
| * ConformerHAR | 5 | 0.613 | 0.612 | 0.611 | 0.499 | 0.482 | 0.348 | 0.533 | 0.491 | 0.483 | 0.612 | 0.602 | 0.601 |
| ¯ SimCLR | 5 | 0.636 | 0.632 | 0.631 | 0.546 | 0.540 | 0.492 | 0.598 | 0.594 | 0.569 | 0.631 | 0.624 | 0.620 |
| ¯ MDC | 5 | 0.742 | 0.744 | 0.741 | 0.706 | 0.704 | 0.707 | 0.728 | 0.716 | 0.708 | 0.703 | 0.672 | 0.678 |
| + FixMatch | 5 | 0.676 | 0.661 | 0.611 | 0.619 | 0.614 | 0.615 | 0.652 | 0.638 | 0.635 | 0.665 | 0.665 | 0.639 |
| + TS2ACT | 5 | 0.835 | 0.810 | 0.811 | 0.752 | 0.745 | 0.747 | 0.773 | 0.760 | 0.756 | 0.780 | 0.764 | 0.763 |
| Ours | 5 | **0.856** | **0.833** | **0.829** | **0.785** | **0.762** | **0.768** | **0.822** | **0.813** | **0.809** | **0.860** | **0.854** | **0.851** |
| * TCN | 10 | 0.748 | 0.721 | 0.720 | 0.532 | 0.522 | 0.515 | 0.689 | 0.630 | 0.616 | 0.612 | 0.583 | 0.563 |
| * ConformerHAR | 10 | 0.795 | 0.774 | 0.775 | 0.668 | 0.589 | 0.576 | 0.731 | 0.716 | 0.705 | 0.712 | 0.718 | 0.696 |
| ¯ SimCLR | 10 | 0.810 | 0.798 | 0.796 | 0.718 | 0.706 | 0.708 | 0.762 | 0.733 | 0.734 | 0.759 | 0.735 | 0.741 |
| ¯ MDC | 10 | 0.825 | 0.834 | 0.832 | 0.749 | 0.735 | 0.740 | 0.732 | 0.722 | 0.725 | 0.806 | 0.791 | 0.795 |
| + FixMatch | 10 | 0.848 | 0.856 | 0.854 | 0.766 | 0.768 | 0.766 | 0.788 | 0.723 | 0.720 | 0.772 | 0.756 | 0.750 |
| + TS2ACT | 10 | 0.916 | 0.914 | 0.912 | 0.822 | 0.814 | 0.813 | 0.874 | **0.869** | 0.852 | 0.901 | 0.892 | 0.899 |
| Ours | 10 | **0.921** | **0.916** | **0.918** | **0.865** | **0.855** | **0.854** | **0.902** | **0.889** | **0.891** | **0.922** | **0.913** | **0.912** |
| * TCN | 20 | 0.821 | 0.811 | 0.807 | 0.719 | 0.720 | 0.684 | 0.792 | 0.753 | 0.749 | 0.721 | 0.675 | 0.667 |
| * ConformerHAR | 20 | 0.845 | 0.843 | 0.841 | 0.745 | 0.688 | 0.652 | 0.833 | 0.819 | 0.802 | 0.772 | 0.760 | 0.752 |
| ¯ SimCLR | 20 | 0.852 | 0.862 | 0.859 | 0.758 | 0.754 | 0.753 | 0.809 | 0.742 | 0.753 | 0.809 | 0.794 | 0.791 |
| ¯ MDC | 20 | 0.864 | 0.859 | 0.858 | 0.755 | 0.756 | 0.757 | 0.768 | 0.751 | 0.755 | 0.846 | 0.832 | 0.833 |
| + FixMatch | 20 | 0.885 | 0.883 | 0.880 | 0.822 | 0.826 | 0.824 | 0.854 | 0.830 | 0.813 | 0.852 | 0.837 | 0.839 |
| + TS2ACT | 20 | 0.924 | 0.922 | 0.921 | 0.887 | 0.886 | 0.888 | 0.926 | 0.917 | 0.916 | 0.943 | 0.936 | 0.941 |
| Ours | 20 | **0.933** | **0.930** | **0.929** | **0.913** | **0.905** | **0.906** | **0.941** | **0.936** | **0.931** | **0.955** | **0.943** | **0.949** |
| * TCN | Fully | 0.923 | 0.921 | 0.920 | 0.915 | 0.915 | 0.913 | 0.909 | 0.891 | 0.876 | 0.935 | 0.926 | 0.923 |
| * ConformerHAR | Fully | **0.965** | **0.963** | **0.963** | 0.942 | 0.931 | 0.932 | 0.943 | 0.939 | 0.935 | 0.964 | 0.952 | 0.957 |
| ¯ SimCLR | Fully | 0.927 | 0.927 | 0.925 | 0.918 | 0.916 | 0.915 | 0.915 | 0.910 | 0.896 | 0.945 | 0.942 | 0.942 |
| ¯ SSL | Fully | 0.906 | 0.897 | 0.895 | 0.822 | 0.797 | 0.786 | 0.915 | 0.898 | 0.901 | NUL | NUL | NUL |
| ¯ MDC | Fully | 0.901 | 0.875 | 0.864 | 0.903 | 0.867 | 0.885 | 0.878 | 0.865 | 0.851 | 0.892 | 0.883 | 0.886 |
| + FixMatch | Fully | 0.924 | 0.923 | 0.922 | 0.914 | 0.908 | 0.906 | 0.913 | 0.912 | 0.912 | 0.942 | 0.935 | 0.933 |
| + TS2ACT | Fully | 0.941 | 0.939 | 0.939 | 0.933 | 0.934 | 0.933 | 0.944 | 0.936 | 0.937 | 0.971 | 0.959 | 0.964 |
| Ours | Fully | 0.958 | 0.946 | 0.947 | **0.952** | **0.946** | **0.942** | **0.967** | **0.962** | **0.963** | **0.979** | **0.967** | **0.968** |

apart. After unsupervised training, we only keep the backbone network and freeze its parameters. We further train a two-layer fully connected network as a classifier using the same amount of labeled samples.

- **Self-supervised Learning (SSL)** [32]. Self-supervised learning predicts time series transformations by training eight proxy tasks. After self-supervised training, they freeze the model parameters and use the labeled data to train a neural network classification head for HAR.

- **Multi-Task Deep Clustering (MDC)** [24]. Multi-task deep clustering uses an autoencoder to get the feature. The K-means clustering algorithm is then applied to divide the dataset into different groups for generating pseudo-labels. Finally, a DNN is further trained for HAR.

## 4.3 Implementation Details

We implement the proposed Vi2ACT using Python and Pytorch. Following the literature [40], we employ 6 Transformer blocks

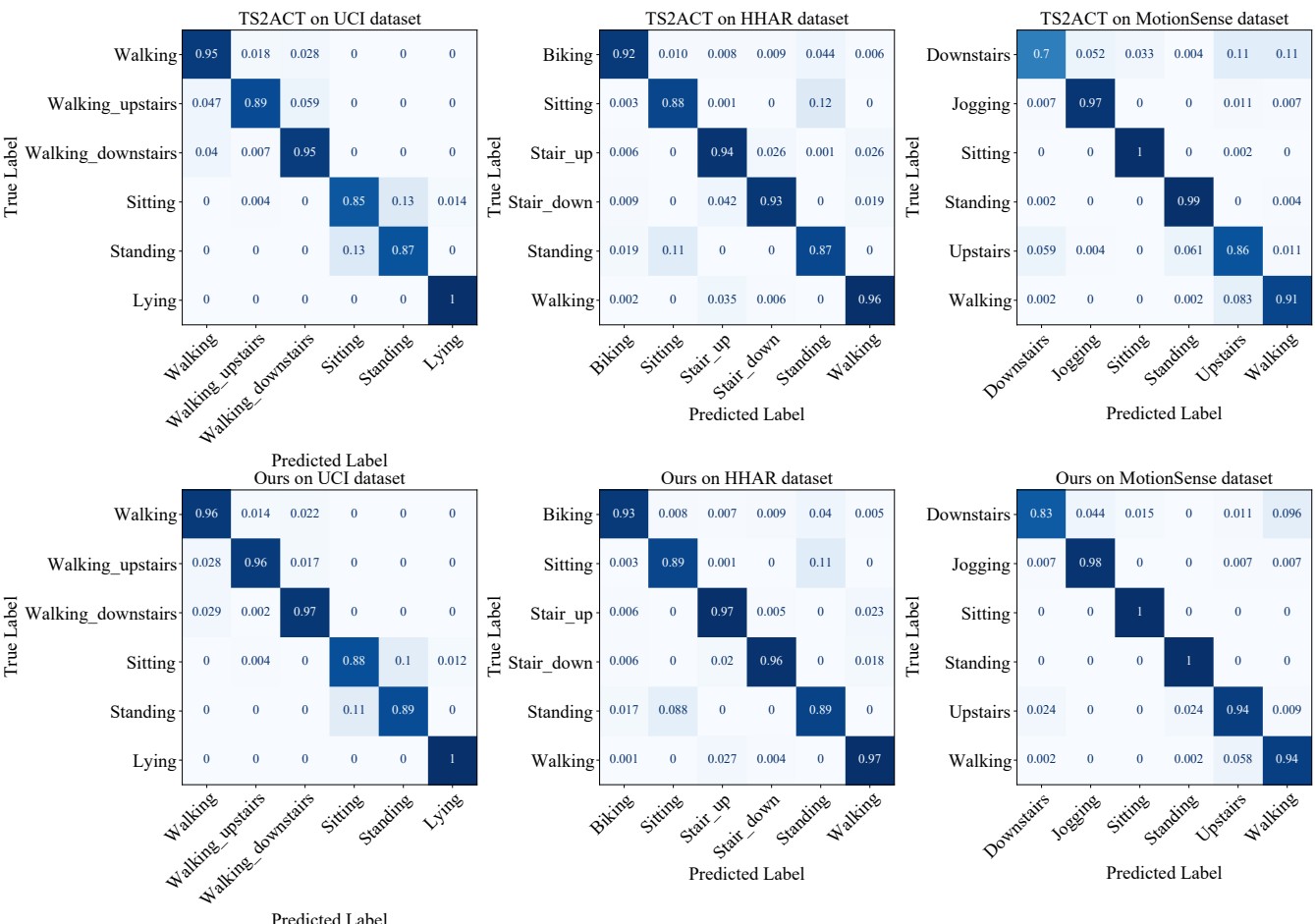

**Figure 3: Confuse matrix of TS2ACT and Our method on different datasets (with 1% labeled samples).**

with the number of heads and the dimension $d$ set to 16 and 512, respectively. We employ the Adam optimizer for both fine-tuning the video encoder and model training stages. During the fine-tuning stage, the learning rate for the video encoder $\mathcal{F}$ is set to 0.000001, while the learning rate for the representation classifier $g$ is set to 0.0005. In the model training stage, the learning rates for the time series encoder $G$ and Representation Conditional Discriminator $D$ are both set to 0.0005. The hyperparameter $\lambda$ for balancing the training of labeled and unlabeled data in Eq. 9 is set to 0.01 by default. In the hyperparameter analysis section, we will provide a detailed discussion of our hyperparameter settings. The experiments were conducted on a personal computer equipped with Intel(R) Core(TM) i7-12700 CPU @4.90 GHz 20 cores, an NVIDIA GeForce RTX 3090Ti graphics card and 32GB RAM.

## 4.4 Comparison with State-of-the-Arts

We compare Vi2ACT with existing work on four commonly used HAR datasets, as shown in Table 2. Vi2ACT exhibits a substantial advantage over other baseline models in all scenarios with limited samples. Even when utilizing the entire set of labeled samples,

competitive results are consistently achieved. Furthermore, Vi2ACT significantly outperforms the other two supervised algorithms, TCN and ConformerHAR. Additionally, Vi2ACT outperforms other semi-supervised and unsupervised representation learning approaches, indicating its superior ability to leverage video representations for enhancing the model's acquisition of semantic representations compared to alternative semi-supervised learning methods. Simultaneously, Vi2ACT addresses the gap between unsupervised pre-training tasks and downstream inference tasks. From the experimental results using fully dataset, Vi2ACT demonstrates its efficacy both in effectively mining semantic representations from temporal data with a limited number of labeled samples and in further enhancing the semantic classification capability of the encoder with a substantial number of labeled samples.

## 4.5 Confusion Matrix

The accuracy for each class is illustrated in the Figure 3. Vi2ACT demonstrates remarkable performance across the three datasets, achieving up to 98% accuracy in specific classes and even reaching 100% accuracy. Furthermore, Vi2ACT exhibits enhanced ability

**Table 3: Performance comparison of aligning representations using different methods**

| Alignment Method | UCI-HAR | | | HHAR | | | MotionSense | | | PAMAP2 | | |
|---|---|---|---|---|---|---|---|---|---|---|---|---|
| | Acc | Recall | F1 | Acc | Recall | F1 | Acc | Recall | F1 | Acc | Recall | F1 |
| Euclidean distance | 0.913 | 0.908 | 0.905 | 0.917 | 0.907 | 0.907 | 0.928 | 0.910 | 0.912 | 0.947 | 0.936 | 0.933 |
| Contrastive Learning | 0.936 | 0.933 | 0.928 | 0.928 | 0.926 | 0.925 | 0.944 | 0.938 | 0.936 | 0.965 | 0.950 | 0.953 |
| Representation Conditional Discriminator | **0.958** | **0.946** | **0.947** | **0.952** | **0.946** | **0.942** | **0.967** | **0.962** | **0.963** | **0.979** | **0.967** | **0.968** |

**Table 4: Performance comparison with different prompt designs, where [class] denotes the class token.**

| Prompts | UCI-HAR | | | HHAR | | | MotionSense | | | PAMAP2 | | |
|---|---|---|---|---|---|---|---|---|---|---|---|---|
| | Acc | Recall | F1 | Acc | Recall | F1 | Acc | Recall | F1 | Acc | Recall | F1 |
| a video of [class] | 0.913 | 0.905 | 0.903 | 0.881 | 0.883 | 0.882 | 0.897 | 0.861 | 0.852 | 0.906 | 0.896 | 0.903 |
| an action video of [class] | 0.931 | 0.928 | 0.926 | 0.934 | 0.920 | 0.921 | 0.923 | 0.916 | 0.918 | 0.937 | 0.922 | 0.921 |
| a man is [class] | **0.958** | **0.946** | **0.947** | **0.952** | **0.946** | **0.942** | **0.967** | **0.962** | **0.963** | **0.979** | **0.967** | **0.968** |

**Table 5: The depth of Representation Conditional Discriminator.**

| # Blocks | Acc | Recall | F1 |
|---|---|---|---|
| 3 | 0.935 | 0.921 | 0.922 |
| 6 | **0.967** | **0.962** | **0.963** |
| 12 | 0.954 | 0.951 | 0.950 |
| 18 | 0.956 | 0.948 | 0.946 |

**Table 6: The width of Representation Conditional Discriminator.**

| HiddenDim | Acc | Recall | F1 |
|---|---|---|---|
| 128 | 0.917 | 0.908 | 0.907 |
| 256 | 0.944 | 0.930 | 0.928 |
| 512 | **0.967** | **0.962** | **0.963** |
| 1024 | 0.963 | 0.961 | 0.961 |

**Table 7: Number of generated videos**

| # Videos | Acc | Recall | F1 |
|---|---|---|---|
| 25 | 0.931 | 0.915 | 0.912 |
| 100 | 0.963 | 0.958 | 0.955 |
| 500 | 0.967 | 0.962 | **0.963** |
| 700 | **0.968** | **0.963** | **0.963** |

**Table 8: Hyperparameter $\lambda$.**

| $\lambda$ | Acc | Recall | F1 |
|---|---|---|---|
| 0.005 | 0.902 | 0.887 | 0.874 |
| 0.01 | **0.967** | **0.962** | **0.963** |
| 0.05 | 0.829 | 0.812 | 0.801 |
| 0.1 | 0.734 | 0.710 | 0.708 |

in discriminating between "Walking downstairs" and "Walking upstairs," thereby effectively mitigating the drawbacks associated with image semantic representation guidance.

### 4.6 Effectiveness of RCD

To validate the effectiveness of our proposed Representation Conditional Discriminator $D$, we compare its performance with directly using the Euclidean distance (i.e., replacing $D$ with mean square error loss function) and contrastive learning (i.e., replacing $D$ with contrastive loss function). The results are summarized in the Table 3. It can be observed that due to the lack of strict one-to-one correspondence between time series and video, our method can more effectively align these two modalities.

### 4.7 Analysis of Different Prompts

We present three prompts templates for Vi2ACT in Table 4, which include "a video of [class] ", "an action video of [class]" and "a man is [class]". We evaluate the model's inference performance across videos generated with different prompts. For each prompt,

we conduct 10 evaluations using all labeled data and compute the average accuracy. The difference in prompt will also lead to performance variance, among which "a man is [class]" has the best performance. Therefore, we use the prompt "a man is [class]" in our model training.

### 4.8 Hyperparameter Analysis

We conduct experiments with different hyperparameters, and the results are shown in Tables 5 6 7 and 8. It is worth noting that a deeper Representation Conditional Discriminator does not necessarily lead to better performance, as excessively deep networks may cause overfitting. Additionally, the performance of Vi2ACT improves as the number of generated videos increases, and the model performance almost converges when using 500 videos. This validates our proposal of using rich video representations to enhance the time series encoder. Finally, setting $\lambda$ to 0.01 achieves a good balance between labeled and unlabeled data.

## 5 CONCLUSION

In this paper, we propose a video-enhanced cross-modal co-learning method, Vi2ACT, to achieve few-shot HAR. We introduce a new data augmentation approach that utilizes a text-to-video generation model to generate class-related videos. Subsequently, a large quantity of video semantic representations are obtained through fine-tuning the video encoder for cross-modal co-learning. Furthermore, to effectively align video semantic representations and time series representations, we design a novel Representation Conditional Discriminator to enhance HAR at the representation level. Finally, we conduct extensive experiments to demonstrate the effectiveness of our method.

## 6 ACKNOWLEDGMENTS

This work was partially supported by the Natural Science Foundation of Jiangsu Province (Grant No. BK20222003), the Collaborative Innovation Center of Novel Software Technology and Industrialization, and the Sino-German Institutes of Social Computing.

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
