# OpenReview forum: "Vi2ACT:Video-enhanced Cross-modal Co-learning with Representation Conditional Discriminator for Few-shot Human Activity Recognition"
_acmmm.org/ACMMM/2024/Conference — MM2024 Poster_

### Official Review · Reviewer_EeJw · 2024-05-19

**Rating:** 5
**Confidence:** 3

**Summary:**

The paper  presents a novel approach for human activity recognition (HAR) using few-shot learning. The authors propose a method called Vi2ACT, which enhances HAR by using a video-enhanced cross-modal collaborative learning approach. This method includes several key components: 1) Text-to-Video Generation. They use a text-to-video generation model to create videos relevant to different activity classes, providing semantically rich video content. 2) Cross-modal Co-learning. The generated videos are used to train a video encoder. This video encoder then collaborates with a time series encoder to enhance the representations used for HAR. 3) Representation Conditional Discriminator. They introduce a novel discriminator that distinguishes whether a representation originates from video data or time series data, helping to align the two modalities more effectively.

**Strengths:**

The authors present a highly novel and theoretically sound approach to enhancing human activity recognition using few-shot learning. Its strengths is as follows:

1) The approach of leveraging video data, rather than just image data, addresses the limitation of temporal information that is often missing in static images, thereby providing a richer dataset for model training.

2) The use of a text-to-video generation model to create class-related videos is an innovative approach in the context of human activity recognition (HAR). This method generates semantically rich video content that supplements the limited labeled sensor data typically available for HAR.

3) The application of conditional GANs to enhance representation learning in HAR is technically sound and leverages the strengths of adversarial learning to improve model robustness and accuracy.

4) The proposed cross-modal co-learning framework, which involves simultaneous training of video and time series encoders, is a novel concept. This method effectively integrates different types of data to enhance the learning process.

**Limitations:**

1. The performance of the model heavily relies on the quality of the videos generated by the text-to-video model. If the generated videos are not of high quality or do not accurately represent the activities, it could negatively impact the model's performance. The paper does not provide a detailed analysis of the quality of the generated videos or how variations in video quality affect the overall performance of the HAR system.

2. The paper does not discuss the computational efficiency of the algorithm. The proposed algorithm incorporates various techniques such as video feature extraction, GANs, and contrastive learning, and we cannot determine how much additional computational overhead this will introduce. Considering that HAR algorithms are often run on lightweight devices, I believe it is essential to evaluate the computational speed.

3. Regarding the third contribution, I believe that extensive experimental validation is a basic requirement for any research work. Unless you have made a special contribution, such as proposing a new dataset or a new evaluation metric, experimental validation should not be considered a core contribution. Writing it this way would appear unprofessional.

**Suitability:**

3

---

### Official Review · Reviewer_UZ37 · 2024-05-20

**Rating:** 4
**Confidence:** 4

**Summary:**

This study propose a video-enhanced cross-modal co-learning method to achieve few-shot HAR. Specifically, the proposed framework applied video semantic representations to enhance the time series encoder for HAR problem.

**Strengths:**

1. The manuscript is easy to understand and follow.

2. The novelty of using video representation to guide time series HAR task is enough.

**Limitations:**

1. Since the proposed method is mainly for cross-modal task, can the authors provide the feature representation visualization results  extracted from multidimensional data to validate the effectiveness of the proposed method?

2. Since this study is focused on few-shot HAR problem, the authors are encouraged to reorganize the related work section. For instance, section 2.1 can be shorten and section 2.2 and 2.3 can be extended. There are several recent few-shot studies [1-3] for HAR problem. In addition, current community start to applied LLMs for crossmodal task [5-6] . The authors are encouraged to select and discuss them in the related work to have a more comprehensive literature review.

[1] Wang, X., Liu, T., Feng, C., Fang, D., & Chen, X. (2023). RF-CM: Cross-modal framework for RF-enabled few-shot human activity recognition. Proceedings of the ACM on Interactive, Mobile, Wearable and Ubiquitous Technologies, 7(1), 1-28.
[2] Wang, X., Yan, Y., Hu, H. M., Li, B., & Wang, H. (2024). Cross-modal Contrastive Learning Network for Few-Shot Action Recognition. IEEE Transactions on Image Processing.
[3] Yuan, H., Chan, S., Creagh, A. P., Tong, C., Acquah, A., Clifton, D. A., & Doherty, A. (2024). Self-supervised learning for human activity recognition using 700,000 person-days of wearable data. npj Digital Medicine, 7(1), 91.
[4] Jain, Y., Tang, C. I., Min, C., Kawsar, F., & Mathur, A. (2022). Collossl: Collaborative self-supervised learning for human activity recognition. Proceedings of the ACM on Interactive, Mobile, Wearable and Ubiquitous Technologies, 6(1), 1-28.
[5] Leng, Z., Bhattacharjee, A., Rajasekhar, H., Zhang, L., Bruda, E., Kwon, H., & Plötz, T. (2024). IMUGPT 2.0: Language-Based Cross Modality Transfer for Sensor-Based Human Activity Recognition. arXiv preprint arXiv:2402.01049.
[6] X. Liu, D. McDuff, G. Kovacs, I. Galatzer-Levy, J. Sunshine, J. Zhan, M.-Z. Poh, S. Liao, P. Di Achille, and S. Patel, “Large
language models are few-shot health learners,” arXiv preprint arXiv:2305.15525, 2023.

3. Format issue. Figure 1 is confused based on the five main components listed in line 230. Please redesign the overall framework for better understanding. In addition, there are some inconsistency in Table 2 regarding bolding formats. Please proofread the manuscript carefully.

**Suitability:**

3

---

### Official Review · Reviewer_S8Be · 2024-05-23

**Rating:** 4
**Confidence:** 3

**Summary:**

This paper presents Vi2ACT, a video-enhanced cross-modal collaborative learning method designed to improve Human Activity Recognition (HAR) in few-shot scenarios. The authors use a text-to-video generation model for data augmentation, generating class-specific videos to enrich the training dataset. Then, the video encoder is fine-tuned to capture video features. Representation Conditional Discriminator is also introduced to distinguish between real and generated videos. The experiments conducted on four major HAR datasets show that Vi2ACT surpasses existing baseline models.

**Strengths:**

- The data augmentation by using video generation models for few-show action recognition is a promising strategy.
- The experiment results in few shot settings outperform the existing methods with a noticeable margin.
- Extensive ablation studies are conducted to showcase the effectiveness of the methods.

**Limitations:**

- As the data augmentation is from pretrained video generator, and the videos are used to train classifier to produce pseudo-label, how did the authors control the quality of the videos?
- In Table 7, the number of generated videos of more than 500 should be examined as well.
- Besides ModelScopeT2V[33], the authors should also examine the effect of video generators.

**Suitability:**

3

---

### Official Review · Reviewer_6Ssf · 2024-05-25

**Rating:** 4
**Confidence:** 3

**Summary:**

This manuscript proposes a method of data augmentation using text-to-video generation models to complete human activity recognition tasks with sensor time-series data. The authors have designed a novel conditional representation discriminator, trained with the idea of conditional generative adversarial networks, to accurately evaluate samples. They demonstrate the superiority of their method compared with several state-of-the-art approaches on four commonly used HAR datasets.

**Strengths:**

1，The method section clearly introduces the overall structure proposed by the authors, with a distinctive idea of training a time-series data encoder using conditional generative adversarial networks.

2，The overall structure of the paper is complete and the narrative is clear.

**Limitations:**

1，It is recommended to check if there is any repetition in the content of the article itself, such as the repeated description of TS2ACT in Section 1 and Section 2.3.

2，Most of the content of the article overlaps with the content of the TS2ACT paper, especially the experimental section, which is almost identical with only minor modifications in the data. It is recommended to rewrite it to reduce the overlap rate.

3，In Section 3.3, it is recommended to indicate that the reference image is Figure 2 in the introduction of the time-series encoder.

4，The discriminator discriminates which input features are time series and which are video generations given the known category c, thereby optimizing the generator training to make the generated features more consistent with category c. Therefore, isn't the overall effectiveness of the entire model particularly susceptible to the quality of the generation model?

5，It is recommended to cite and introduce the specific text-to-video generation model used in the paper.

**Suitability:**

3

---

### Meta-Review · Area_Chair_cW5D · 2024-06-27

**Recommendation:** Accept (Poster)
**Confidence:** 5

**Metareview:**

All reviewers think the paper has proposed a novel method, where using t2v model to create data is innovative. Reviewers have some concerns on the quality of the generated videos, and some details in experiments. After the author’s rebuttal, most of the concerns have been resolved. The AC recommend accept for this submission.